# Standard Block and Modular Dwelling Designs in Hong Kong's Public Housing

Lu Wang [1,2,*], Jingru Cyan Cheng [2], Wojciech Mazan [2] and Sam Jacoby [2]

1    State Key Laboratory of Subtropical Building Science, School of Architecture, South China University of Technology, Guangzhou 510640, China

2    School of Architecture, Royal College of Art, London SW7 2EU, UK

\*    Correspondence: wanglu_scut@outlook.com

**Abstract:** This paper examines the role of standard block and modular dwelling designs in Hong Kong's public housing provision since the mid-1950s. It explores how standard types have evolved in relation to housing policies, demographic and socio-economic changes, and minimum space requirements. In contrast to other countries, Hong Kong lacks defined space or room standards. In the absence of space standards, Hong Kong relies on a living density standard. This paper studies the historical development of Hong Kong's public housing in terms of dwelling size as a measure of housing quality, questioning the effectiveness of standard block and dwelling designs as housing design controls and highlighting the contextual nature of dwelling usability and size. The analysis is based on public housing design projects, policies, and data implemented or presented by the Hong Kong government, particularly the Hong Kong Housing Authority.

**Keywords:** public housing; standard block design; Hong Kong

## 1. Introduction: Public Housing in Hong Kong

Compared to many other housing systems, such as that in the United Kingdom, Hong Kong lacks defined space or room standards for dwellings. Space standards are commonly used regulatory instruments to control the minimum size of newly built homes, especially in the subsidised housing sector that includes social, public, and affordable housing [1]. They derive from normative ideas about household compositions and dwelling use, incorporating standard furniture schedules or dimensions, activity zones, and access and circulation areas to determine minimum room dimensions and floor areas. As a result, space standards reflect social norms and cultural expectations regarding how homes are used [2] and have become widely accepted as an easily measurable assessment of dwelling usability and housing quality [1,3,4]. This connection between housing quality and dwelling usability is supported by research on how the availability of space in a home can directly impact the health and well-being of its occupants, an awareness heightened by the COVID-19 lockdowns [5–8].

In the absence of space standards for controlling housing outcomes, Hong Kong has utilised a living density standard, but only for public rental housing allocation, based on a minimum internal floor area (IFA) per person and measured by median rent–income ratio (MRIR) limits [9]. While this indicates a concern for overcrowding, it also demonstrates limited regulatory interest in assessing dwelling usability, on which space standards are based. Housing in Hong Kong is therefore characterised by its compact size, as evident from the average living space of only 13.6 $m^2$ per person in public rental housing (PRH) for an average household size of 2.7 persons, with 47.1% of dwellings having a floor area ranging from only 30 $m^2$ to 39.9 $m^2$ and another 36.5% being even smaller than this [10]. Due to limited space, flats are often rented out as "shells" with wall partitions provided only around the bathroom and kitchen.

However, to ensure consistent housing outcomes, most public housing in Hong Kong was built using standard block types starting in the 1950s until the 2000s and, more recently, by adopting the modular flat design (MFD) approach. The historical exclusive reliance on standard designs to regulate housing outcomes, without other measurable standards, is relatively unique to the People's Republic of China and Hong Kong, where a predominance in the public housing supply meant that extensive formal regulation was unnecessary, unlike in most other countries, where minimum standards for dwelling and room sizes are prescribed [1]. The paper is, therefore, interested in understanding the effectiveness of standard block and dwelling designs in improving housing quality over time, a regulatory approach that has been possible in the public housing context of Hong Kong as it is distinct from others in several ways.

As is well known, population growth and limited land supply has contributed to Hong Kong's housing market becoming the least affordable in the world. The real price level of housing doubled from 2008 to 2018 [11]. The average housing price in 2019 was HKD 9.7 million (GBP 982,000) [12]. The median multiple of a house price divided by household income reached 18.8 in 2022 [13]. This lack of affordability has made very small dwelling sizes acceptable to the market but compelled the government to play a major role in the supply of public housing. While the government's involvement in housing supply is strengthened by state ownership of all land, revenue from land premiums paid by private developers also constitutes a large source of fiscal income. The acute shortage of land poses a complex economic, political, and social challenge in allocating land for private and public housing developments [14].

As a consequence of these pressures, the Hong Kong Housing Authority, a government agency, is by far the main public housing supplier and thus has been able to exert nearly complete control over its design standardisation. Given this comprehensive government control over public housing, it has been legally exempt from the requirements of the Buildings Ordinance that apply to private housing developments. The scale of public housing provision is also substantial within the overall housing market. According to the Housing Bureau [10], 45.7% of the population lived in public permanent housing in 2021. This included 30% living in rental housing and 15.7% in subsidised sale flats. Despite the significant scale of public housing, a persistent housing shortage in Hong Kong continues.

While the use of standard block and flat designs ensures cost-effective construction and consistent housing outcomes, it does not necessarily guarantee high housing standards or effective design control when housing suppliers and housing needs diversify. The extensive standardisation of housing tends to predominantly cater to the needs of the most common household types and lifestyles, neglecting the marginalised needs of groups such as the ageing population or non-traditional households. This is exerting growing pressure on Hong Kong's housing market. Likewise, the recent shift toward a public housing supply model involving the private sector has increased pressure for greater formal regulation of public housing design.

This paper, therefore, explores how Hong Kong's public housing has historically performed regarding dwelling size as a measure of housing quality. This paper asks the following: To what extent are standard block and dwelling designs effective means of housing design controls, and when are additional or different design regulations needed? Given the small sizes of dwellings in Hong Kong, the paper discusses how this dwelling usability is contextual to a place and time.

The paper is based on the analysis of public housing design projects, policies, and data implemented or presented by the Hong Kong government, especially the Hong Kong Housing Authority in its different guises.

## 2. Standard Block and Dwelling Designs

Different standardised housing block and dwelling designs in Hong Kong characterise four distinct periods that encapsulate the evolving role of public housing [15] and changes in minimum dwelling size (Figure 1).

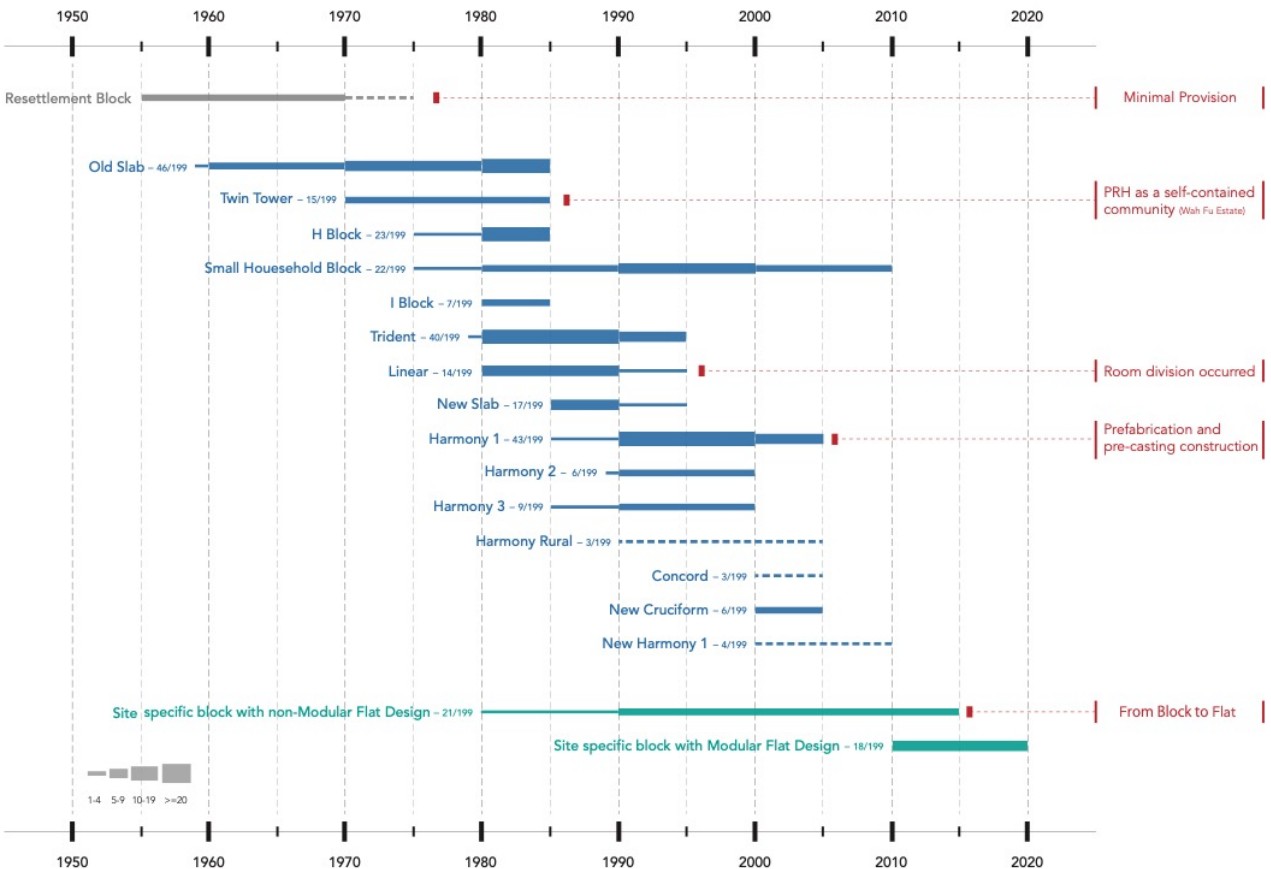

**Figure 1.** Timeline of standard block types and site-specific blocks with modular flat design, including the number of developments using them (out of a total of 199).

The first period, spanning from 1954 to 1973, was defined by basic housing provisions. Standard block designs featured single-room units with a small WC and kitchen forming a service core located on a balcony.

The second period, from 1973 to 1987, witnessed a shift towards a more comprehensive housing supply. Single-room layouts were replaced by units with internal room divisions. The WC and kitchen moved into the interior of the flat, resulting in smaller balconies that eventually disappeared.

During the third period, from 1987 to 2002, public housing began to include subsidised sale flats in addition to rental homes, but the distinct standard block organization and flat layouts were retained.

In the current fourth period, since 2002, standardisation at the block scale has been abandoned, with an increasing focus on modular flat design for all housing tenures, offering greater flexibility to adapt to different site conditions and user requirements.

*2.1. Basic Public Housing Provision (1954–1973)*

The influx of migrants from mainland China during the Chinese Civil War (1945–1949) led to a population surge from 600,000 in 1945 to 2 million in 1951 [16]. Consequently, in the late 1940s, the British Colonial Office commissioned Patrick Abercrombie to devise a long-term development plan for Hong Kong. Abercrombie's Preliminary Planning Report recommended dispersing the population concentrated around the harbour by creating new residential districts and satellite towns in the hinterland of Kowloon and the New Territories [17]. In the 1950s to 1960s, Hong Kong's population further doubled from 2 million in 1951 to 3.9 million in 1971 [15,18], creating an immense housing demand.

However, it was not until 53,000 people were made homeless by the Shek Kip Mei Fire in 1953 that concerns about safety and sanitation prompted the British colonial government to take action and abandon its laissez faire housing approach [19]. A Resettlement Department was created in 1954, tasked with the construction and management of new housing under the Resettlement Programme [20]. The emergency housing for the fire victims was the first public housing built in Hong Kong.

The Mark I type resettlement housing block, designed by G. P. Norton, an architect of the Public Works Department, is considered the prototype of Hong Kong's public housing. It combined elements of the local tenement house (*tong lau*) with post-war working-class terraced housing in Britain [21]. In use from 1954 to 1964, the H-shaped Mark I block had six or seven storeys and consisted of two parallel slabs (Figure 2). Each slab had back-to-back single-room units accessed from an external deck. The emergency housing units were only 11.15 m² (120 ft²) in size. Recognising the substandard nature of these dwellings, the potential conversion into self-contained flats was considered already at the inception of the housing model (Figure 3).

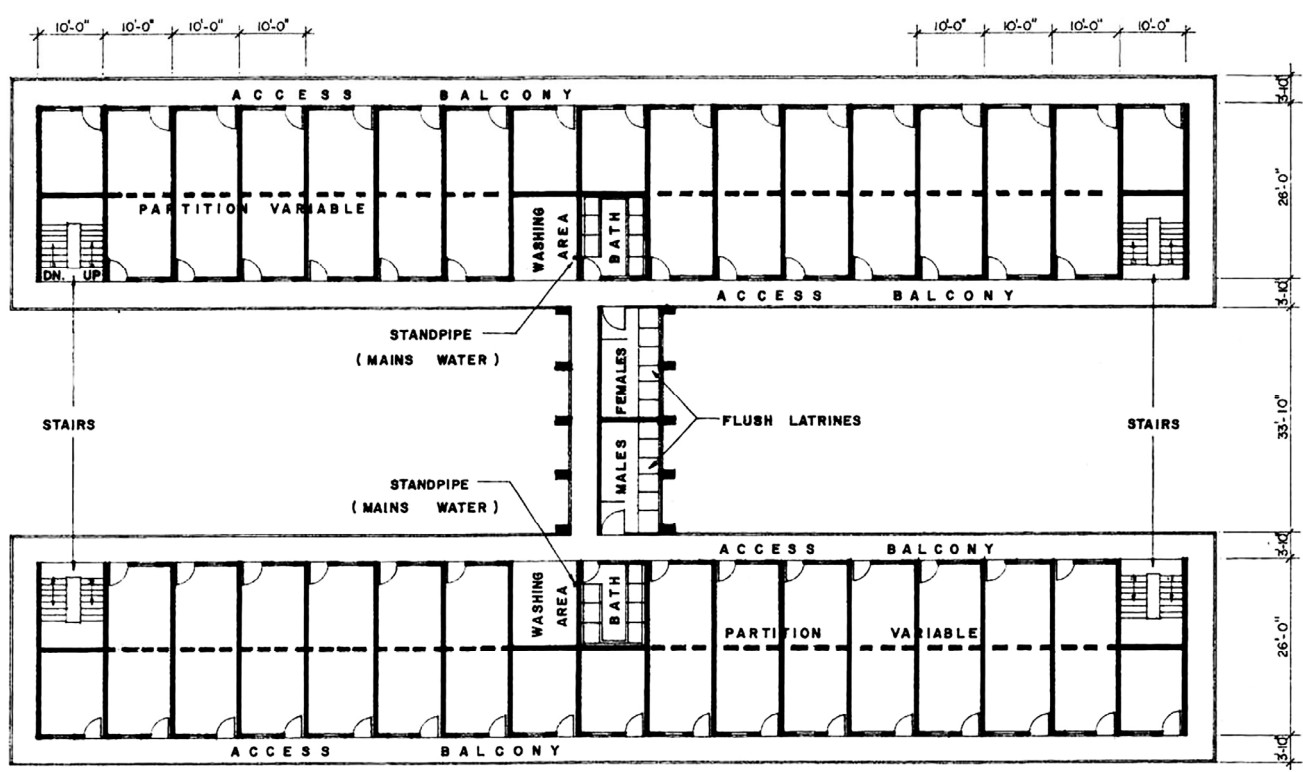

**Figure 2.** Mark I typical resettlement block plan. Source: Commissioner for Resettlement, Annual Departmental Report, financial year 1954–55. Hong Kong: Government Printer (1955) [22].

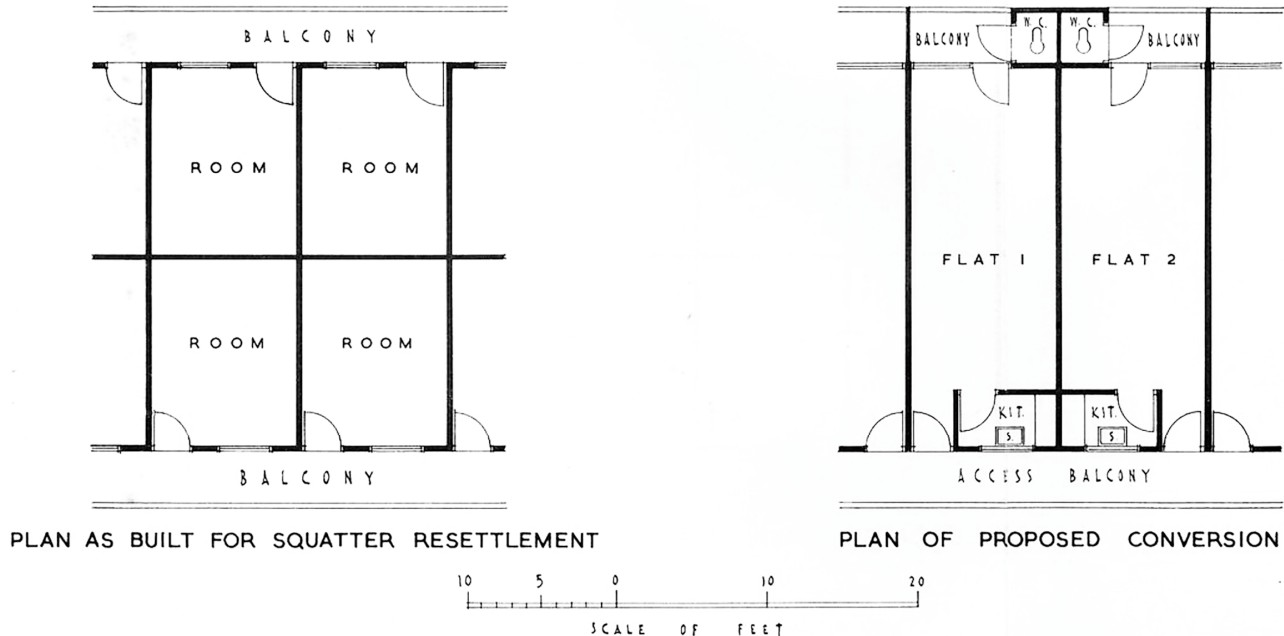

**Figure 3.** Method of future conversion of Mark I resettlement building into self-contained flats. Source: Commissioner for Resettlement, Annual Departmental Report, financial year 1954–55. Hong Kong: Government Printer (1955) [22].

However, despite these immediate concerns with dwelling size, the Mark II (1961–1964) block only provided larger communal services areas in the space linking the two slabs, added vertical and horizontal circulation, and created internal courtyards by enclosing the two open ends of the 'H'. Just the units at the end of the slabs, used to rehouse local land-owning villagers, were larger at 28.8 m$^2$ and had their own water supply, kitchen, and balcony [23].

Only in 1964, ten years into the resettlement programme, did the government's Review of Policies for Squatter Control, Resettlement, and Government Low-cost Housing acknowledge the need for improving living conditions by increasing the standards of resettlement housing. Subsequently, all dwelling units were required to have a private balcony, WC, and water supply.

Starting with the Mark III type in 1964, dwellings were organised around a double-loaded corridor, allowing each unit to have a private balcony. A year later, the Mark IV introduced a private WC and water supply on the balcony, creating the first self-contained public housing units. With the provision of electricity and running water in each unit, the cooking space was moved inside the dwelling, often on or near the balcony, effectively transforming it into a service area. The later Mark V and VI block types also began to offer a range of unit sizes for different family sizes. As a result, the Mark VI block, first appearing in 1970, had a standard unit size of 13 m$^2$ (140 ft$^2$) for a family of four.

To meet growing housing demand, the height of buildings increased from the eight-storey tall Mark III to the sixteen-storey Mark IV, V, and VI blocks. The Mark IV block types were also the first public housing with lifts. However, the increased density necessitated the provision of additional common facilities and better social and welfare services, with subsequent Mark blocks arranged to create semi-open courtyards that could be used for communal and commercial neighbourhood facilities.

During the first two decades of public housing in Hong Kong, alongside the provision of resettlement housing, larger flats for low- and mid-income families were built by the non-governmental and non-profit Hong Kong Housing Society and the "former" Hong Kong Housing Authority (which was restructured in 1973 to become part of today's Housing Authority).

The first public housing development created by the former Housing Authority, the North Point Estate (1957) designed by Eric Cumine, targeted low-income white-collar workers. The estate comprised eleven-storey tall towers that provided 1955 self-contained flats, accommodating a population of over 12,300. Offering four main flat types and their variations, the estate catered to households with three to eight family members. Each unit had a kitchen, WC, and balcony, along with a smaller utility balcony for storage and drying clothes. Unlike other public housing estates at the time, the North Point flats were constructed with all partition walls installed.

Planned as a community, the estate offered a comprehensive range of social and recreational facilities accessible to both residents and the general public. The amenities included a primary school, workshops, clinics, a pharmacy, a post office, a community centre, and shops [24]. The layout and provision of a residential neighbourhood with integrated public facilities set a design norm for public housing estates in the coming years [25].

Until 1973, the former Housing Authority developed a total of ten public housing estates. Apart from the North Point Estate and So Uk Estate (1963), these estates employed, at least in parts, the so-called Old Slab block configuration (Figure 4). Similar to the later IV to VI Mark block designs, the typical dwelling unit of the Old Slab featured an open living–sleeping space without partitions, along with a "utility" balcony with a WC and water supply. The floor plan was organised around a double-loaded corridor with a central vertical circulation core connecting two wings. The Old Slab blocks were either eight or fifteen storeys tall, with lift access provided to every third floor, and remained popular until the mid-1980s.

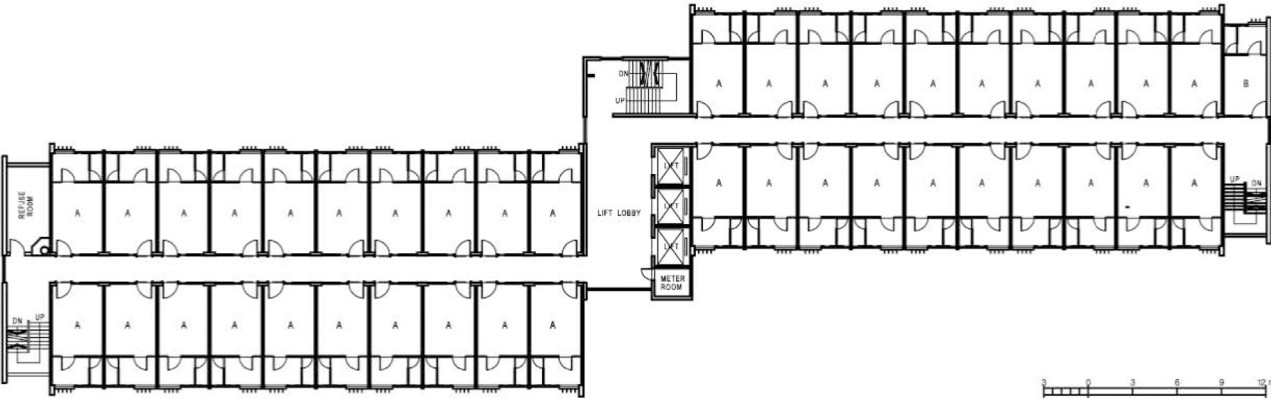

**Figure 4.** Old Slab typical standard block floor plan with staggered 180-degree connection. The connection angle could be varied to suit different site configurations. Source: Hong Kong Housing Authority. Available online: https://www.housingauthority.gov.hk (accessed on 5 December 2023) [26].

Based on a new town concept, the Wah Fu Estate supplied 9100 flats for 54,000 residents, making it the largest estate created by the former Housing Authority. Phase I (1967) with 12 Old Slab blocks provided flats ranging from 28.3 to 36.5 m$^2$. Phase II (1970) featured six Twin Tower blocks (Figure 5), comprising a 20-storey tower and a 23-storey tower interconnected by a shared vertical circulation core, with four different flat types from 36 m$^2$ to 46 m$^2$ arranged along a single-loaded corridor open to a central atrium space. This layout allowed for ventilation, natural light, and increased security, as residents on the same floor could see each other's doorways. In comparison to the Old Slab block, the units in the Twin Tower block, built throughout the 1970s and early 1980s, included an additional kitchen space connected to the utility balcony.

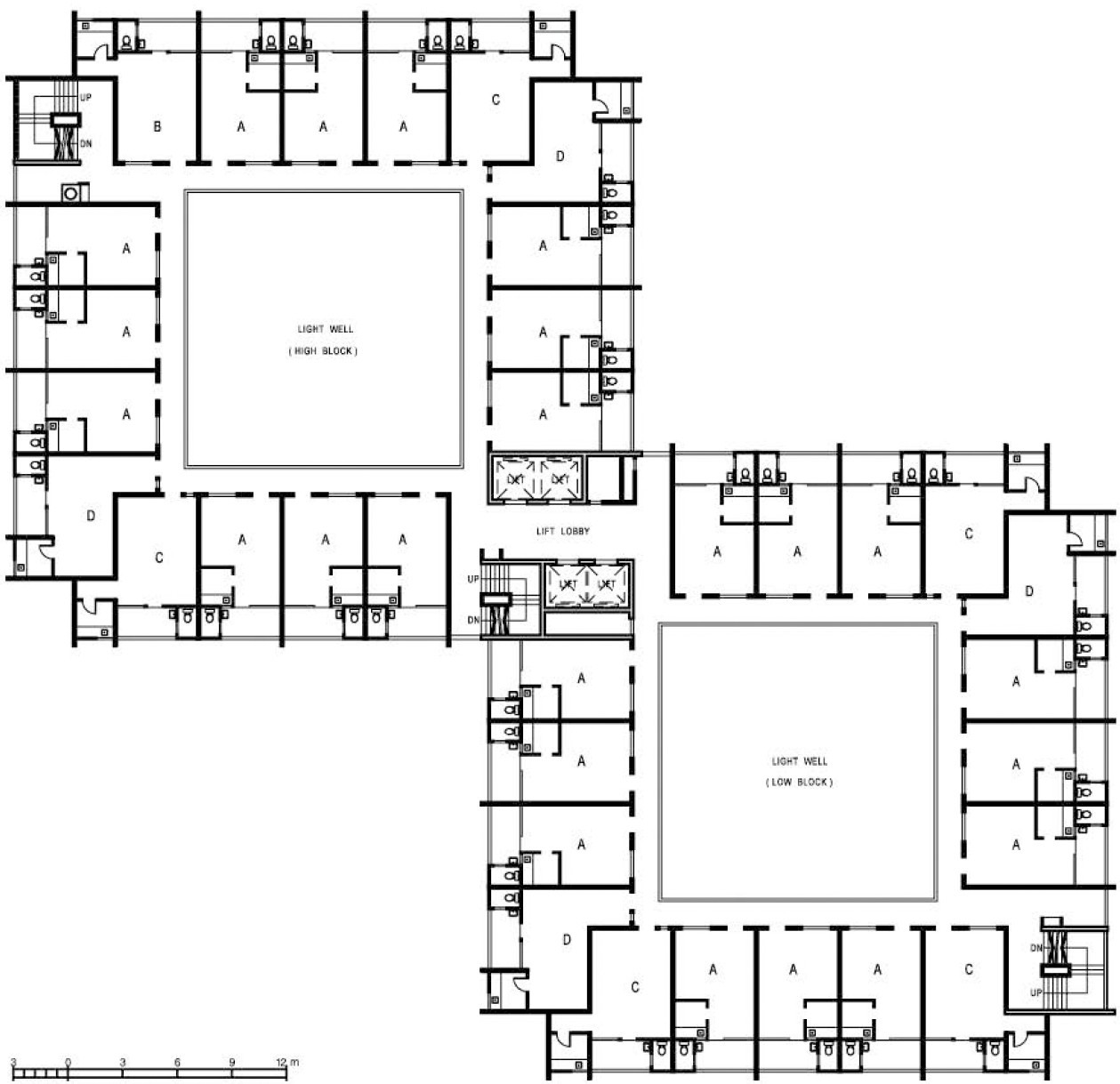

**Figure 5.** Twin Tower typical standard block floor plan. Source: Hong Kong Housing Authority. Available online: https://www.housingauthority.gov.hk (accessed on 5 December 2023) [26].

By the end of the Resettlement Programme in 1973, over 1 million people had been (re)housed [27]. While the average household size in the 1960s was around 4.5 persons, the standard units in resettlement blocks were designed for families of five [15], with the typical monthly rent for a five-person family being HKD 18–38 [28]. Since the housing targeted fire victims and squatters, no income limit for eligibility was set. However, the basic housing standard had improved only insignificantly, from 2.22 m$^2$ (24 ft$^2$) per person in the Mark I housing in the 1950s to 3.25 m$^2$ (35 ft$^2$) by 1968 in the Mark V and VI block types (Table 1). Even the larger housing created by the former Housing Authority only marginally improved housing standards, with the North Point Estate providing an average space of just over 3.7 m$^2$ (40 ft$^2$) per person [29].

**Table 1.** Comparison of standard block designs from the 1950s to 1970s.

| | | Mark I | Mark II | Mark III | Mark IV | Mark V | Mark VI | Old Slab | Twin Tower |
|---|---|---|---|---|---|---|---|---|---|
| **General** | **Period** | 1954–1964 | 1961–1964 | since 1964 | since 1965 | | since 1970 | 1960s–1980s | 1970s–1980s |
| | **Programme** | Resettlement estate (1954–1973) | | | | | | PRH/TPS | PRH |
| | **Av. household size** | 1961: 4.4 | | | | 1966: 4.7 | 1971: 4.5 | | |
| **Dwelling** | **Person** | 5 | | 4–6 | | | 4–8 | | |
| | **Size (m$^2$)** | 11.15 | 11.15 or 28.8 | 14–21 | 14–21 | 14–27 | | | 36–46 |
| | **No. of types** | 1 | 2 | 2 | 3 | 4 | 3 | 2 | 4 |
| | **Av. living space** | Minimum 2.23 m$^2$/person | | | | | Minimum 3.25 m$^2$/person | | |
| | **Partitioning** | Single room, no partitions | | | | | | | |
| | **Features** | | | Balcony, electricity | Self-contained unit, balcony, electricity, water | | | | (plus cross-ventilation) |
| **Block** | **Access** | Deck access | | Double-loaded corridor | | | | | Single-loaded |
| | **Units per floor** | 2 slabs: 60 | 2 slabs: 68 | ca. 40 | ca. 43 | ca. 48 | ca. 58 | 2 slabs: 41–44 | 34 |
| | **No. of floors** | 6–7 | 7–8 | 8 | 16 | | | 8 or 15 | 20 or 23 |

Notes: PRH: public rental housing; TPS: tenant purchase scheme.

The housing provided by both the Housing Society and the Housing Authority targeted lower-middle-income families that earned between HKD 500 (GBP 45.5) and HKD 1250 per month. The average living space offered was 3.25 m$^2$ per adult, excluding kitchens, WCs, and balconies [30]. The Housing Society had a monthly household income limit for eligibility set at HKD 1000 (up to HKD 1250 for higher rent units). The standard monthly rent for a five-person family ranged from HKD 42 to HKD 110. On the other hand, the Housing Authority had an income limit set between HKD 400 and HKD 900 (up to HKD 1250 for higher rent units), with standard rents ranging from HKD 60 to HKD 120 [15]. In comparison, the Government Low-Cost Housing Programme targeted lower-income families with a monthly household income limit of HKD 500 and offered standard rents from HKD 40 to HKD 60 [15]. The average living space provided was also 3.25 m$^2$ per adult, including a private balcony that could accommodate a cooking bench and had a water tap.

*2.2. Comprehensive Provision (1973–1987)*

Industrialisation and the improvement of living standards exerted pressure for better housing. At the same time, advancements in construction technology during the 1980s led to taller buildings and higher population densities.

The initial Buildings Ordinance, enacted in 1955 to regulate private-sector housing developments, transformed the prevailing housing typology from low-rise tenements to the podium-tower design, emblematic of high-density housing in Hong Kong. By the late 1960s, new private-sector housing developments required a minimum rent of HKD 200 per month for viability [16]. However, the 1971 Census showed that 59% of the population had a monthly household income below HKD 800 (equal to a rent-to-income ratio of 25%) and 70% below HKD 1000 (equal to a rent-to-income ratio of 20%) [18]. This rendered private housing increasingly unaffordable, necessitating the construction of public housing at larger scales and higher densities to meet growing demand.

The Leftist Riots, starting in 1967, included calls for better social welfare provisions and housing conditions [31]. In response, the Ten-year Housing Programme was launched in 1971, with the housing estates provided by the programme focusing on the integration of public spaces, facilities, and amenities, as well as landscaping. The small footprint of high-rise buildings left 70–80% of the site available for potential use for public and communal services [20]. The emphasis on the neighbourhood dimension of estates has been a key planning principle of housing policy since the 1970s.

Starting in 1973, public rental housing (PRH) programmes served as the primary procurement model to supply heavily subsidised housing to low-income groups. While the PRH estates in the 1970s initially used the Old Slab and Twin Tower block designs, a new H-Block type was introduced in 1976 and widely used, especially throughout the first half of the 1980s. The H-Block could accommodate up to 15 flats on each floor, interconnected by a central corridor and a vertical circulation core, rising up to 27 floors. Single H-Blocks could be connected at the ends of their wings to form a double H-Block, the most commonly used type (Figure 6).

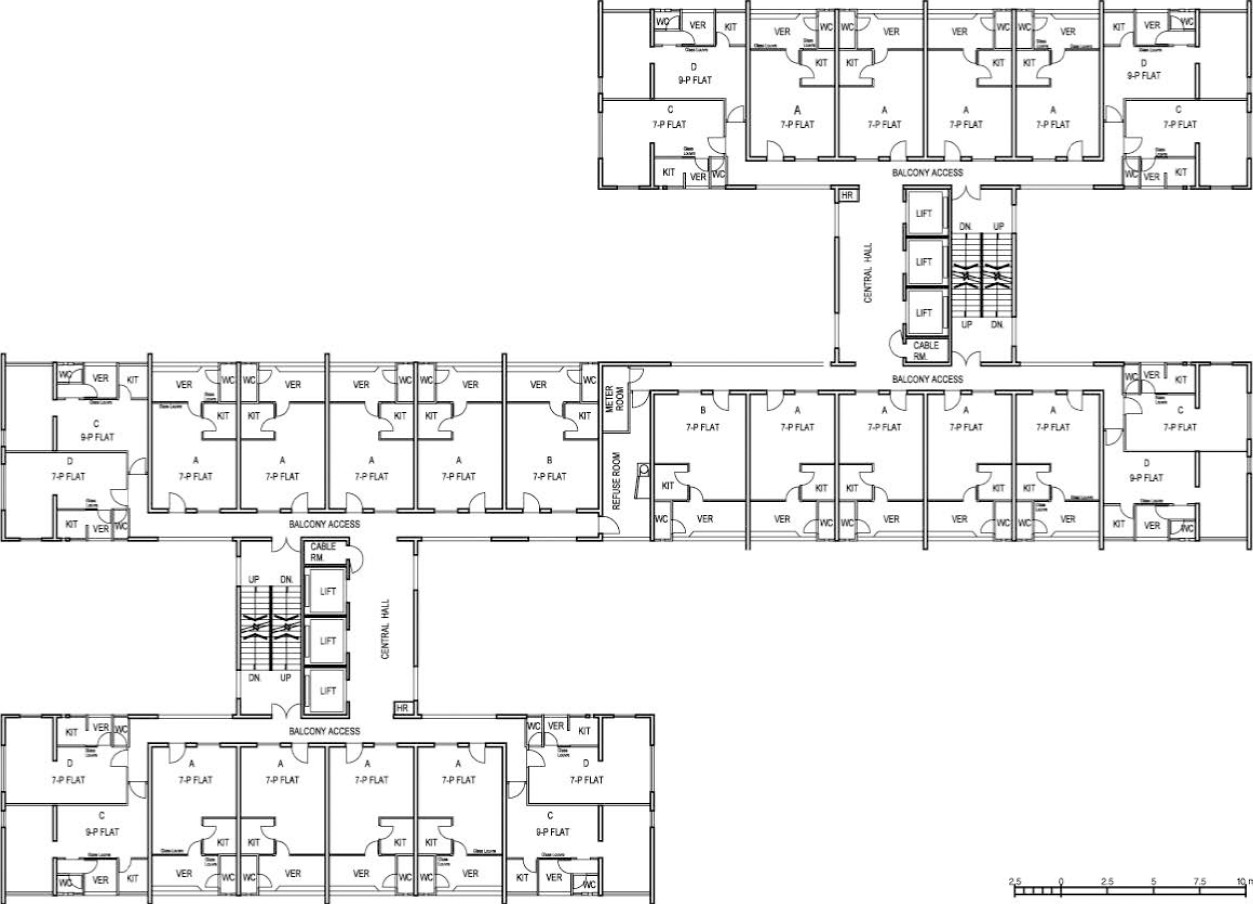

**Figure 6.** Double H typical standard block floor plan. Source: Hong Kong Housing Authority. Available online: https://www.housingauthority.gov.hk (accessed on 5 December 2023) [26].

Other new public housing types introduced in the 1980s were the I-Block, New Slab, and Linear block, with their variations specifically designed for the constraints in narrow urban sites. While the I-Block and New Slab had a central double-loaded corridor, the Linear type featured a single-loaded corridor, enabling greater variations in flat types (Figure 7). The Linear type was extensively used in redevelopment programmes in Kowloon and New Kowloon and similar in length to the Mark I and II blocks.

The 1980s also saw the emergence of the highly popular Trident block (Figure 8), primarily used for public rental housing estates in new towns. This marked a significant transition from corridor-based linear block designs to high-rise towers in public housing, reaching up to 35 floors. The Y-shaped plans of the Trident blocks increased privacy for residents, as the flats on the three wings were no longer directly facing each other. In addition, multi-room dwelling layouts provided each room with direct access to the exterior through a so-called "re-entrant bay" window, improving natural lighting and ventilation.

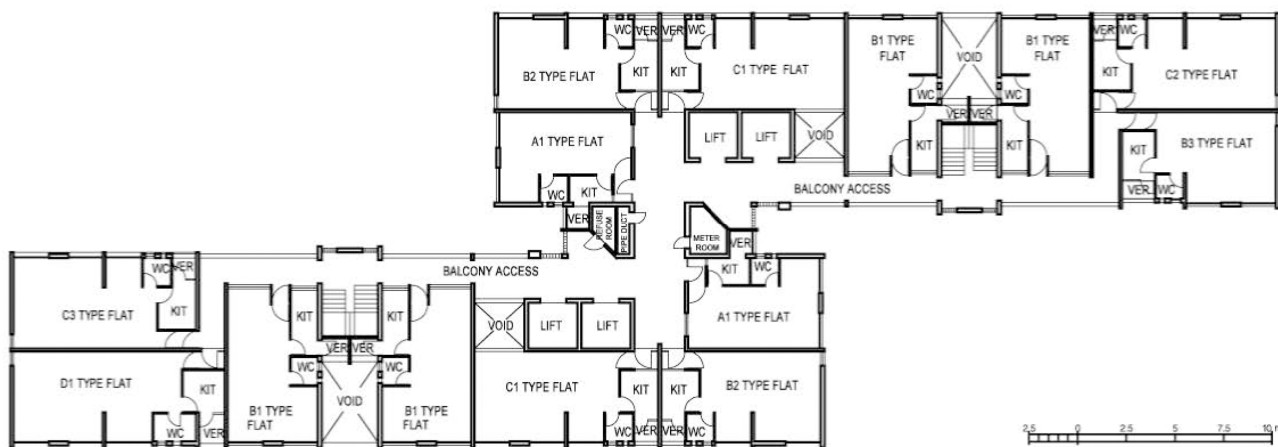

**Figure 7.** Linear typical standard block floor plan. Source: Hong Kong Housing Authority. Available online: https://www.housingauthority.gov.hk (accessed on 5 December 2023) [26].

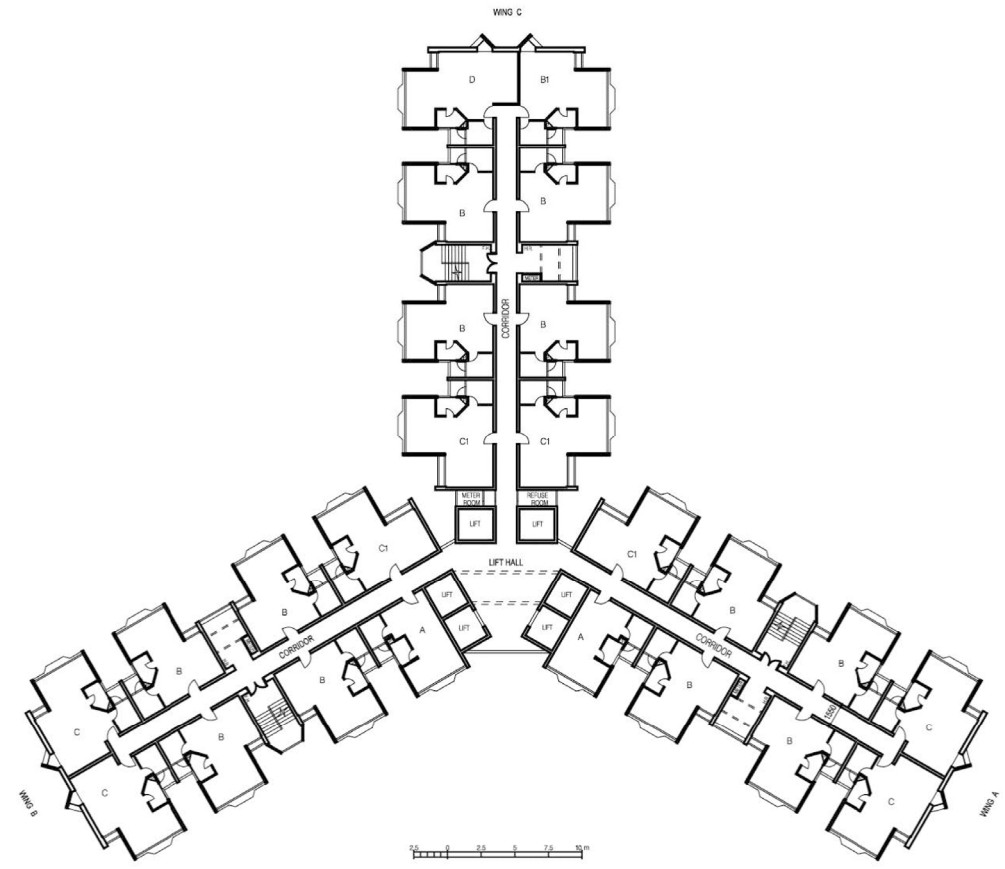

**Figure 8.** Trident 3 typical standard block floor plan. Source: Hong Kong Housing Authority. Available online: https://www.housingauthority.gov.hk (accessed on 5 December 2023) [26].

The Ten-year Housing Programme, despite falling significantly short of its target of 1.8 million dwellings, provided new homes for approximately 1 million residents. This shortfall was primarily attributed to the economic recession following the oil crisis of 1974 and the existing public housing structure, with public housing bodies reorganised in 1973 into the new Hong Kong Housing Authority. Nevertheless, by 1981, public housing accommodated 2 million people, equal to 38.5% of Hong Kong's population. During the period from 1971 to 1981, the proportion of the population living in new towns also increased by

nearly 9% [32], reflecting an effort to relocate residents from overcrowded urban areas and establish a local labour force near the industrial centres in the New Territories.

The new public housing not only led to a doubling of average floor areas to a minimum of 7 m$^2$ per person in the Trident housing (Table 2) but, importantly, also embraced a more holistic approach to housing and living quality. This involved developing housing estates with the necessary public facilities, services, and amenities to function as more autonomous neighbourhoods.

**Table 2.** Comparison of standard block designs from the 1970s and 1980s.

| | | H Block | Household | I Block | New Slab | Linear | Trident I | Trident II | Trident III | Trident IV |
|---|---|---|---|---|---|---|---|---|---|---|
| General | Period | 1976–1980s | 1980s–2000s | mid-1980s | 1980s–1990s | 1980s | 1980s–mid-1990s | | | |
| | Programme | PRH/TPS | | PRH | | | PRH/TPS | | | |
| | Av. household size | 1981: 3.9 | | 1986: 3.7 | | | | | 1991: 3.4 | |
| Dwelling | Person (P) or bedroom (B) | 1–6 | 1/2P; 2/3P; 1B; 2B | | | 4–6P | 3–4P | 4–6P | | |
| | Size (m$^2$) | 40 | IFA 17.6; 22.9; 31.7; 41.8 | | | 36–55 | 28 | 35–49 | | |
| | No. of types | 4 | | 2 | 1 | 4 | 1 | 2 | 4 | 5 |
| | Av. living space | Minimum 3.25 | Minimum 4.25 m$^2$/person | | | | Minimum 7 m$^2$/person | | | |
| | Partitioning | | Single room, no partitions | | | | Multi-room, self-partitioning | | | |
| | Features | Self-contained unit, balcony | Self-contained unit, some for elderly with larger kitchen | Self-contained unit | Self-contained unit, no balcony | Self-contained unit, cross-ventilation, no balcony | Self-contained unit, no balcony | Self-contained unit, separate window to all rooms | | |
| Block | Access | Single-loaded | Site-specific block design | Double-loaded corridor | | Single-loaded | Central lift core, double-loaded corridor | | | |
| | Units per floor | 15 | | 27 | 26 | 2 slabs: 14 | 36 | | 24 | 18 |
| | No. of floors | 27 | | 21 | | 27 | | 35 | | |

Notes: PRH: public rental housing; TPS: tenant purchase scheme; IFA: internal floor area.

Just prior to the launch of the Ten-year Housing Programme in 1971, the median monthly rent in public housing in Hong Kong was HKD 41, and the overall median monthly household income was HKD 708. By 1981, these figures had risen to HKD 151 for rent and HKD 2995 for household incomes [32]. Public housing rents were markedly lower, ranging from 20% to 30% of those for comparable accommodation in the private sector. Consequently, the rent-to-income ratio in public housing was around 5%, as opposed to 16% in the private sector [33]. Throughout the Ten-year Housing Programme, the average waiting time for housing allocation was on average 8 to 9 years in urban estates and 3 to 4 years for those in new towns.

*2.3. Home Ownership (1987–2002)*

The global promotion of homeownership and privatisation of housing in the 1980s, along with a soaring property market, resulted in the construction of numerous large-scale private housing estates in Hong Kong. Policy changes in the 1980s and 1990s also led to a shift from public rental flats to homeownership as the preferred long-term housing tenure, with the government encouraging greater private-sector involvement.

The Housing Authority launched its first Home Ownership Scheme (HOS) in 1976 to develop public housing for sale at below-market prices. This initiative targeted public

rental housing (PRH) tenants and lower-middle-income families—whose income exceeded the PRH threshold but fell short of affording housing on the private market [34].

The HOS began with the Sui Wo Court in Sha Tin (1980) development in the New Territories, consisting of two types of two-bedroom flats and one type of three-bedroom flat, ranging from 43 m$^2$ to 65 m$^2$ in gross floor area (GFA). Like private housing, all flats featured a separate living–dining room, kitchen, bathroom, and two or three bedrooms.

The first Long Term Housing Strategy of Hong Kong, announced in 1987, formally introduced a drive to privatise public housing [19]. In an attempt to reduce public expenditure, privatisation came along with new policies to limit housing subsidies, encourage public housing tenants to buy homes, and deregulate the housing market [35].

To address the more affluent tenants using public housing resources, the Housing Subsidy Policy was introduced in 1987. It made households living in PRH for 10 years or more, with an income twice the Waiting List Income Limit, pay double the standard rent. This income limit, surject to annual review, varied according to household size. For instance, in 1989, it was set at HKD 3000 for a single person and HKD 9400 for a family of ten or more [33]. This aimed to encourage better-off tenants to vacate public housing, with their rents raised to largely align with those in the private sector. In 1996, the Policy on Safeguarding Rational Allocation of Public Housing Resources was established, marking a significant shift by considering asset values when determining eligibility for housing subsidies. Households paying double rent were mandated to declare their net assets every two years, and if they exceeded the Waiting List Income Limit by 110 times, they had to pay market rent [36].

The HOS created a need for new standard blocks in the 1980s, such as the Cruciform, Flexi, and Windwill types. Although HOS flats had layouts similar to private high-rise housing, their size ranged from 37 to 60 m$^2$, comparable to PRH dwellings. However, whereas PRH flats typically had a service core and a living–sleeping space that tenants had to divide themselves, HOS flats provided separate rooms for different functions and typically two bedrooms, with their layout also increasingly determined by the optimisation of natural ventilation and lighting.

Following studies by the Housing Department, the Housing Authority developed the modular Harmony block in 1988 (Figure 9). A variant of the Cruciform type, it was the last standard block design created by the Housing Authority before the transition to non-standard blocks in the early 2000s. Harmony types were widely used in both public rental housing and Home Ownership Scheme developments throughout the 1990s. With a total of 10 sub-variants, Harmony 1 is the most commonly employed standard block design in Hong Kong, having been employed in 43 out of 199 PRH estates by 2019. The Harmony standard block also introduced an integrated modular approach to gain greater flexibility and efficiency in design and construction processes. Harmony types are based on five standard unit sizes: a one-person (1P) unit (ca. 17 m$^2$ for 1–2 persons), a one-bedroom unit (ca. 35 m$^2$ for 3–4 persons), a two-bedroom unit (ca. 43 m$^2$ for 4–5 persons), and two three-bedroom units (ca. 50 m$^2$ for 5–7 persons) [37].

In 1991, single-person households accounted for 14.3%, two-person households 18.4%, three-person households 19.4%, four-person households 22.7%, and five-person households 14.3%, with an average household size of 3.4 [38]. The new HOS units provided an average living space of 7 to 10 m$^2$ per person, significantly higher than the previous standards of 4.25 m$^2$ in the 1980s, 3.25 m$^2$ in the 1970s, and 2.23 m$^2$ in the 1960s and 1950s [17]. Notably, kitchens were enlarged to accommodate domestic appliances, and three-bedroom units offered separate WC and shower rooms. Following the implementation of the Harmony designs, public housing units no longer included open balconies to prevent tenants from enclosing them to increase the floor area [39].

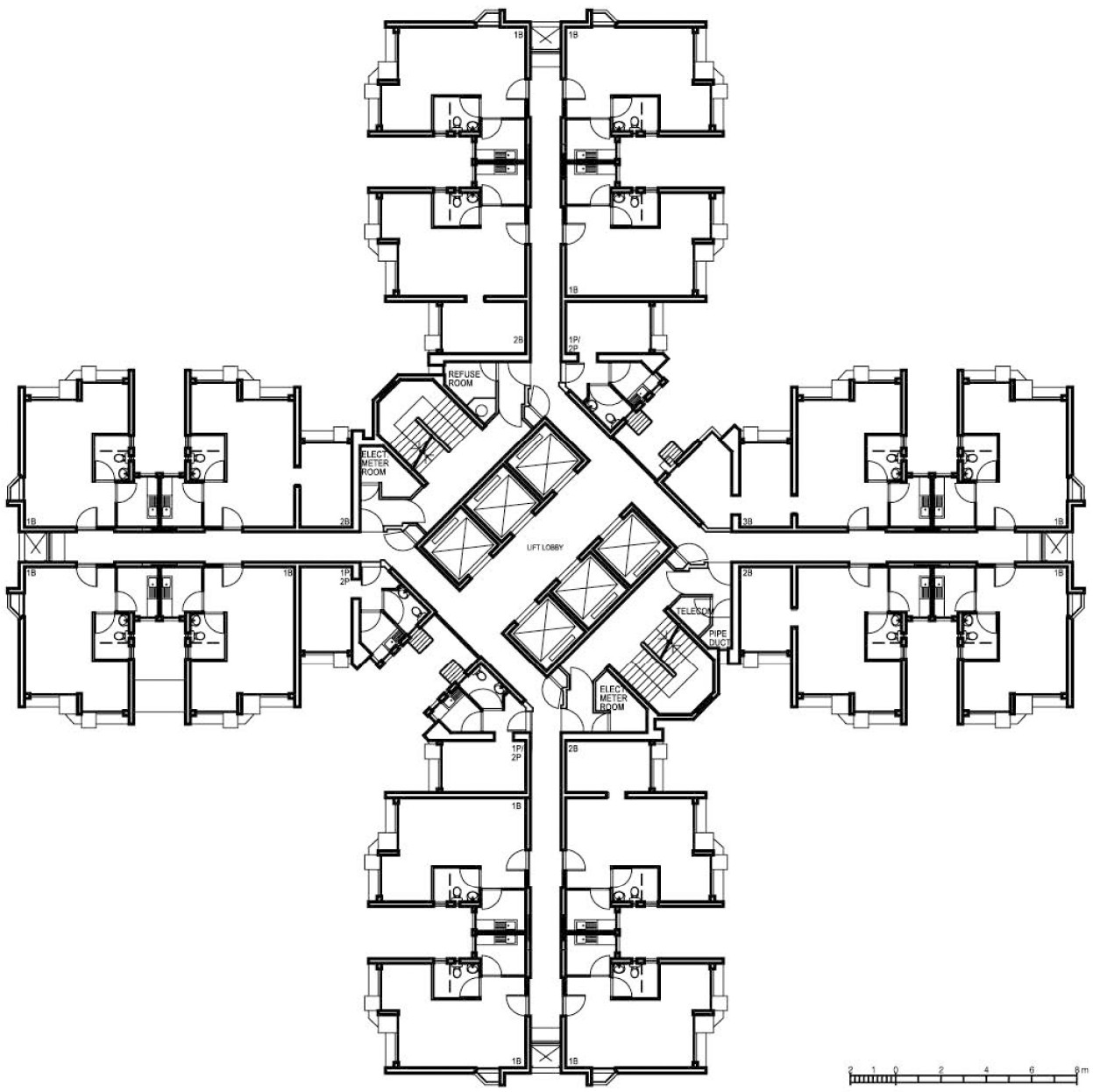

**Figure 9.** Harmony 1 (Option 1) typical standard block floor plan. Source: Hong Kong Housing Authority. Available online: https://www.housingauthority.gov.hk (accessed on 5 December 2023) [26].

Starting in 1992, the PRH stock was reclassified using categories based on flat sizes that aligned with the new Harmony block provision [40]. The categories were small flats with an internal floor area (IFA) of 13–25 m², one-bedroom flats with an IFA of 32–43 m², two-bedroom flats with an IFA of 39–48 m², and three-bedroom flats with an IFA of 49–60 m². The distribution of these flat types across the PRH stock significantly changed over time. In 1993/94, it was 11% for small flats, 39% for one-bedroom flats, 44% for two-bedroom flats, and 6% for three-bedroom flats, but by 2002/03, it changed to 25%, 24%, 34%, and 17% [41–45]. Therefore, one- and two-bedroom homes became the primary forms of housing provided by the PRH programmes (Table 3).

**Table 3.** Comparison of standard block designs from the 1980s to 2000s.

| | | Cruciform A | Cruciform B | Flexi 1–3 | Windwill | | New Cruciform | Harmony | Concord |
|---|---|---|---|---|---|---|---|---|---|
| General | Period | 1976–1980s | | | | 1980s–mid-2000s | | 1988–mid-2000s | 1990s–mid-2000s |
| | Programme | HOS | | | | PSPS | HOS | PRH/HOS | HOS |
| | Av. household size | | | | | 1991: 3.4 | 1996: 3.3 | 2001: 3.1 | |
| Dwelling | Person (P) or bedroom (B) | 3B | 2B | 2–3B | 1–2B | | 1–3B | 1P unit/1–2P; 1B unit/2–3P; 2B unit/3–4P; 3B/5–7P | 2B; 3B |
| | Size (m²) | SA 43–52 | | SA 38–52 | SA 37–60 | SA 34–80 | SA 37–63 | 17; 35; 43; 52 | SA 47–61 |
| | No. of types | 1 | 2 | 6 | 3 | | 3 | 2 each | 2 |
| | Av. living space | Minimum 7 m², actual up to ca. 10 m²/person | | | | | | | |
| | Partitioning | Separate rooms | | | | | | Multi-room, self-partitioning | Separate rooms |
| | Features | Self-contained unit with separate windows to all rooms | | | | | | | |
| Block | Access | Central core (no corridor) | | | | | Central lift core, double-loaded corridor | | Central core |
| | Units per floor No. of floors | 8 | | 16 | 6–10 | | 10 | 16 or 18<br>27; 37; 39 | 8 |

Notes: HOS: home ownership scheme; PSPS: private sector participation scheme; PRH: public rental housing; SA: saleable area.

### 2.4. Modular Design (2002–)

The 1997 Asian financial crisis led to a collapse of the property market and in 2002 to the suspension of government-led homeownership programmes in Hong Kong. The reliance on private-sector housing caused extensive problems of housing affordability and access, forcing the government to reverse its policies in the 2010s and not only resume but expand subsidised homeownership schemes. This decision was in part driven by demographic and lifestyle changes that had created new housing challenges such as homes for an ageing population [46] or young adults and graduates.

For almost fifty years, the Hong Kong Housing Authority regulated the design and supply of public housing using standard block types. However, limited development sites for public housing necessitated a more flexible design approach capable of responding to individual site constraints for greater land utilisation [47]. The Housing Authority, therefore, abandoned standard block designs and started to explore site-specific housing block designs in the 2000s. Completed in 2007, Phase 1 of the Shek Kip Mei redevelopment was the first project completed using this approach.

In parallel, the government experimented with modular flat designs, revisiting design principles first explored in 1988 with the Harmony types. By 2008, the Housing Authority had developed four flat prototypes with a variation each, which became the first eight modular flat designs (MFDs) for public rental housing. The four flat types corresponded to public housing categories and sizes based on target occupancy rates: 1P/2P flat (IFA 14.1–14.5 m²), 2P/3P flat (21.4–22 m²), 1B flat (30.2–31 m²), and a 2B flat (37–38 m²) [48]. The MFD types have become the predominant model for public housing in Hong Kong, regardless of tenure. Despite this standardisation, site-specific modifications are possible, for example, for corner flats, to maximise developments.

While in the 2000s the minimum allocation standard for public rental housing was 7 m² IFA per person, the actual rate provided by PRH was around 13 m². This increase was mainly due to larger HOS flats being reallocated for PRH use after the suspension of the HOS in 2002 [47]. The average living space per person in the new MFD homes remained within this range. The MFDs from 2008 were updated in 2013 (Figure 10), 2015, and 2018. While the overall flat layouts remained largely the same, the main changes related to technical improvements of the designs.

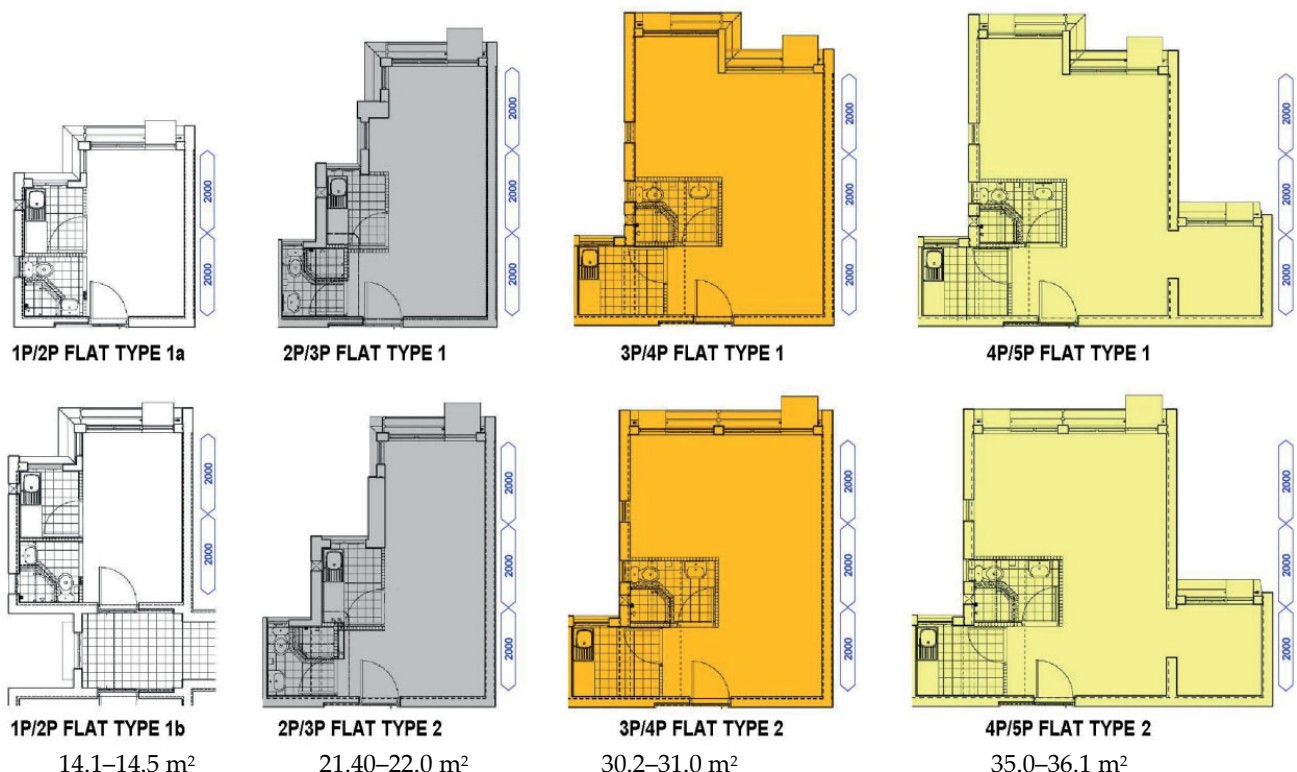

**Figure 10.** Modular flat design, 2013/2015 version. Source: Legislative Council Panel on Housing, Modular Flat Design for public housing development of the Hong Kong Housing Authority (2013) [49].

In 2015, the MFD types were officially renamed and reclassified to create greater standardisation across both PRH and HOS flats. Subsidised sale flats are measured by their saleable areas (SAs) like private housing, with HOS flats subdivided into a Class A (under 40 m$^2$) and Class B (between 40 and 69.9 m$^2$), creating the following approximate correlation between IFA and SA:

- Type A (1P/2P): IFA 14.1–14.5 m$^2$, SA around 17–17.5 m$^2$—Class A;
- Type B (2P/3P): IFA 21.4–22.0 m$^2$, SA around 26–26.5 m$^2$—Class A;
- Type C (3P/4P): IFA 30.2–31.0 m$^2$, SA around 35–36 m$^2$—Class A;
- Type D (4P/5P): IFA 35–36.1 m$^2$, SA around 41–45 m$^2$—Class B;
- Type E (5P or more): IFA 45 m$^2$, SA around 52–53 m$^2$—Class B.

In modular flat designs, the main distinction between developments for public rental or subsidised sale schemes lies in their flat combinations, which are used to manage the mix of housing provision (Table 4). This difference is evident both in floor plans and the distribution of flat types.

Complementary to the modular flat design, a Modular Integrated Construction (MIC) approach was introduced in 2017 to promote the use of "freestanding volumetric modules (with finishes, fixtures, fittings, etc.) manufactured off-site and then transported to site for assembly" [48]. This new housing design focus means that the reliance on private housing developers and construction specialists is again growing. As the government is retreating from its near full control over public housing design and supply, a greater level of housing regulation is required, raising new questions for the Housing Authority and housing stakeholders.

**Table 4.** Comparison of modular flat designs (MFDs) since 2000.

| | | MFD 2008 | MFD 2011 | MFD 2015 | |
|---|---|---|---|---|---|
| General | Programme | PRH | | PRH/HOS | |
| | Av. household size | 2006: 3 | 2011: 2.9 | 2016: 2.8 | |
| Dwelling | Person (P) or bedroom (B) types | 1–2P; 2–3P; 1B; 2B | | A: 1–2; B: 2–3P; C: 3–4P; D: 4–5P | E: 5 + P |
| | Size (m$^2$) | 14–14.5; 21.4–22; 30.2–31; 37–38 | 14–14.5; 21.4–22; 30.2–31; 35.36.1 | | 45 |
| | No. of types | 2 each | 2; 2; 3; 3 | 2 each | 1 |
| | Av. living space | Minimum 7 m$^2$, actual up to ca. 13 m$^2$/person (<5.5 m$^2$ = overcrowded) | | | |
| | Partitioning | Multi-room, self-partitioning | | | |
| | Amenities, services | Self-contained unit with separate windows to all rooms (since 2013: universal design) | | | |
| Block | Access and floors | Site-specific block design | | | |

Notes: PRH: public rental housing; HOS: home ownership scheme.

With the resumption of the Home Ownership Scheme, different eligibility criteria were announced by the Housing Authority before the launch of each sale exercise. In 2019, HOS flats were priced 52% below the market rate. The eligibility criteria for that year required the household gross monthly income to be below HKD 60,000 (GBP 6100) and assets value below HKD 1.96 million (GBP 198,000), while the median monthly household income was HKD 28,700 (GBP 2900) [50]. To support less well-off households, the Green Form Subsidised Home Ownership Scheme (GSH), initially trialled in 2016 and fully implemented in 2018, offers homes for sale at 10% above HOS sale rates. Although the GSH is a new scheme, the estates are converted PRH estates. In addition, the HOS Secondary Market was not only resumed but also extended to White Form status buyers (low- to middle-income families not currently residing in PRH) through the White Form Secondary Market Scheme (WSM) that came into effect in 2018.

## 3. Dwelling Size and Use

The minimum dwelling size in public rental housing is determined by the number of occupants and the living space and living density standards established by the Housing Authority. Since its inception in the 1950s, the living density standard has undergone significant changes, arguably in recognition of changes in dwelling use and household composition.

### 3.1. Dwelling Size

Until 1973, living density standards for public housing were based on a distinction between Group A Estates (government low-cost housing and Housing Authority estates) and Group B Estates (former resettlement estates) [51]. For Group A, the minimum standard was set at 3.25 m$^2$ net living area per adult, with children under 10 years old counted as half a person. For Group B, the minimum standard was 2.23 m$^2$ of net living area per adult.

In November 1973, the Housing Authority removed the division between the two groups and introduced a universal minimum standard of 3.25 m$^2$ net living area per person. This was followed by almost a decade (1973–1982) of continuous increase in the net living area, which reached an average of 4.43 m$^2$ per person by the end of 1982. In the same year, the Housing Authority decided to lower the living density standard below the average, at 4 m$^2$ of net living area per person (or 5.5 m$^2$ of the IFA per person) for new housing estates. But in 1987, it was raised again to 4.2 m$^2$, while the IFA remained unchanged.

In 1991, the Housing Authority introduced a new standard to define dwelling floor areas based on 'affordability', measured as a ratio between the median rent and income of prospective residents. The provision of space was divided into two groups, 5.5 m$^2$ IFA per person for the median rent–income ratio (MRIR) of 15% and 7 m$^2$ IFA per person for the MRIR of 18.5%. A review of the dual standard allocation by the Housing Authority in

1992 concluded that the IFA provision per person should remain unchanged, and the 1991 standards are still in use today.

The current MFD IFAs are all below 45 m$^2$, and in 2020, more than 80% of the public rental housing stock had an IFA of less than 40 m$^2$ [52]. While the size of public rental flats significantly increased from the 1950s to the 1990s, there has been a downward trend since. Specifically, the multi-room rental flats of the Harmony types in the 1990s largely provided dwellings with an IFA of 32–48 m$^2$ (one- and two-bedroom units), whereas the most common Type C flat of the MFD has an IFA of around only 30 m$^2$ (one-bedroom).

But private housing remains very small too. According to a study in 1999, the average saleable area of private flats in Hong Kong was 15.6 m$^2$ per person, whereas the minimum space standard per person was 21.34 m$^2$ in the UK and 18 m$^2$ in Japan [53]. Yet, even in London, the average floor space per person in 2015–2017 was in fact 33 m$^2$ [54]. This confirms that during the 1990s, the average dwelling space in private housing was larger than that in the public sector.

However, it is important to note that this higher average saleable area includes high-end housing in its calculation [55]. In 2018, it was estimated that 66% of all public housing flats were smaller than 39.9 m$^2$, and the same was true for 45% of private homes by the end of 2019 [56]. This claim is supported by data showing that by 2020, HOS flats with a size between 40 and 60 m$^2$ made up almost 70% of the provision [52], with more than half of HOS housing completed between 2018 and 2020 being below 40 m$^2$ in size [56]. A recent report by Our Hong Kong Foundation in 2021 further projects that by 2024, private flats will reach a record low of less than 45 m$^2$, two-thirds of their average size in 2012 [57].

While private-sector housing is subject to development controls, there are no regulations governing its size, which has resulted in the construction of so-called nano apartments. What the comparison of public and private sector housing shows is that basic standards such as the living density standards are both effective and necessary to safeguard minimum dwelling sizes. If left unregulated, the private housing sector fails to supply affordable and decent housing.

### 3.2. Dwelling Interior and Use

An important aspect of public housing in Hong Kong is a process known as "tenant fitout". Instead of providing residents with fully fitted-out flats, the Housing Authority typically provides unfinished "shells". This makes it the responsibility of the residents to construct and finish the interior walls and floors.

The most common choice is to subdivide a "shell" into either small bedrooms or a large "common room" for living, dining, and other shared activities. Residents prioritise the maximum number of bedrooms over their size as well as the size of the common room. Nevertheless, the size, layout, and uses of these spaces have undergone significant changes as spatial standards have improved [58].

Due to the limited space available in public housing units, a diverse and efficient utilisation of space by different families or households is essential (Figure 11). Popular strategies to make flats more useable are foldable furniture, tatami mats, and the creation of a mezzanine level if the ceiling height permits. For example, in the case of older public housing, it was common to create a mezzanine sleeping area, whereas in newer housing with lower ceilings, the use of foldable furniture is common.

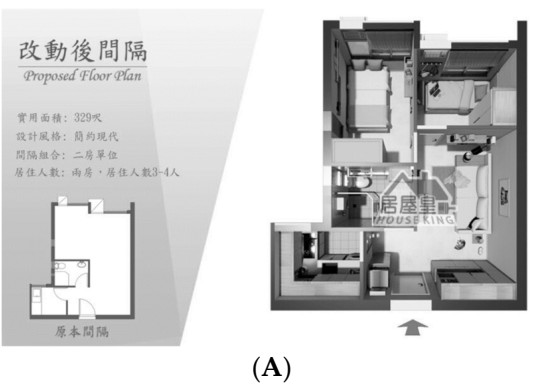
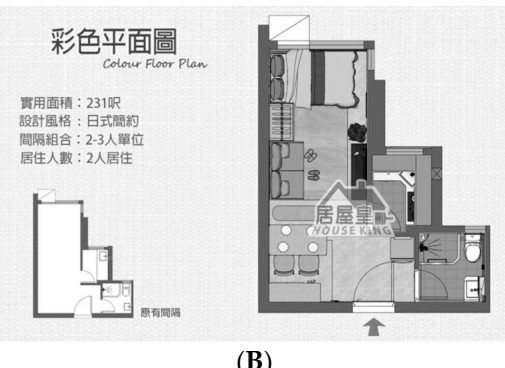

(**A**)  (**B**)

**Figure 11.** Diverse fitout and use of public housing by different families. (**A**) A 329 ft² unit for a 3/4-person family. (**B**) A 231 ft² unit for a 2-person family. Source: Public Housing with 2–3 Persons, Case-1. Available online: https://houseking.com.hk/ (accessed on 7 December 2023) [59].

Innovative ways of fitting out small public housing flats in Hong Kong are shown in the television programme Gou Si Qi Ze. In one example, a 25 m² (232 ft²) flat for a two-person family utilises two-sided wardrobes and foldable furniture to maximise the use of space (Figure 12). The total budget of the project was HKD 185,000, which is comparable to the government's estimated cost of HKD 150,000–200,000 for the basic fitout of a public housing flat of 300–400 ft² [60].

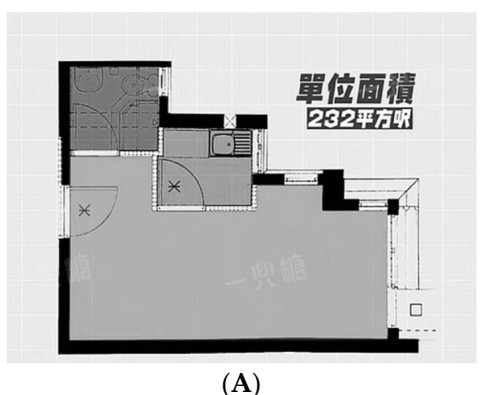

(**A**)

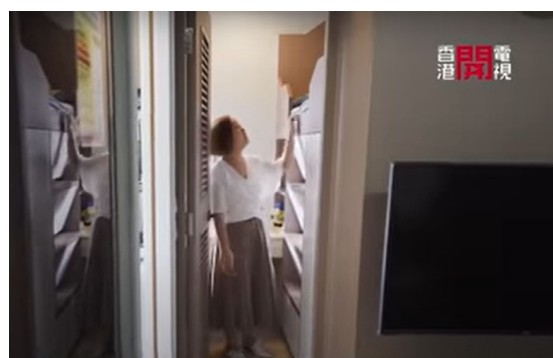

(**B**)

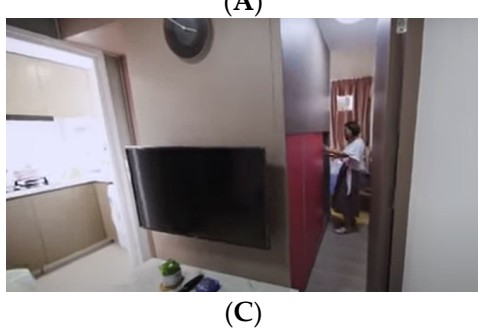

(**C**)

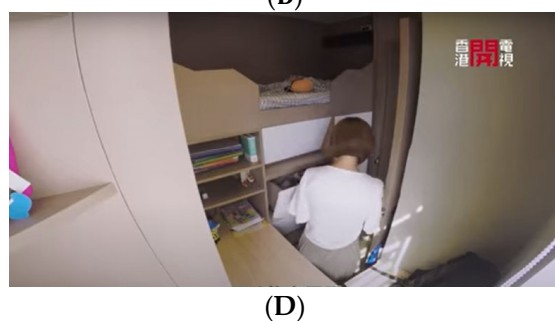

(**D**)

**Figure 12.** Furniture instead of wall partitions is used to separate public housing into different functional zones. (**A**) Layout of a 25 m² flat. (**B**) Use of two-sided wardrobe to separate bedroom spaces. (**C**) One side of the wardrobe and large bedroom. (**D**) Small bedroom with cupboard for children. Source: EP9-2 Interior Design of Small Units. Available online: https://www.youtube.com/watch?v=FaAXgq5iVqM. (accessed on 7 December 2023) [59].

Due to the cost and small size of private-market nano flats (often around 18.6 m² or 200 ft²), public housing is desirable to lower- and mid-income groups. In 2023, the monthly rent for a 15.2 m² (164 ft²) nano flat in North Point reached HKD 10,800 [61], whereas the rent for a comparable public housing unit was only around HKD 1600.

Similar to public housing, private nano flats require innovative interior design and use. For example, an episode of the Gou Si Qi Ze series shows how a 15 m² flat is fitted out to accommodate a four-person family, including parents and two children. A new mezzanine level was created to provide a new sleeping area for the parents and one of the children, while a foldable bed was used for the other child and for guests (Figure 13).

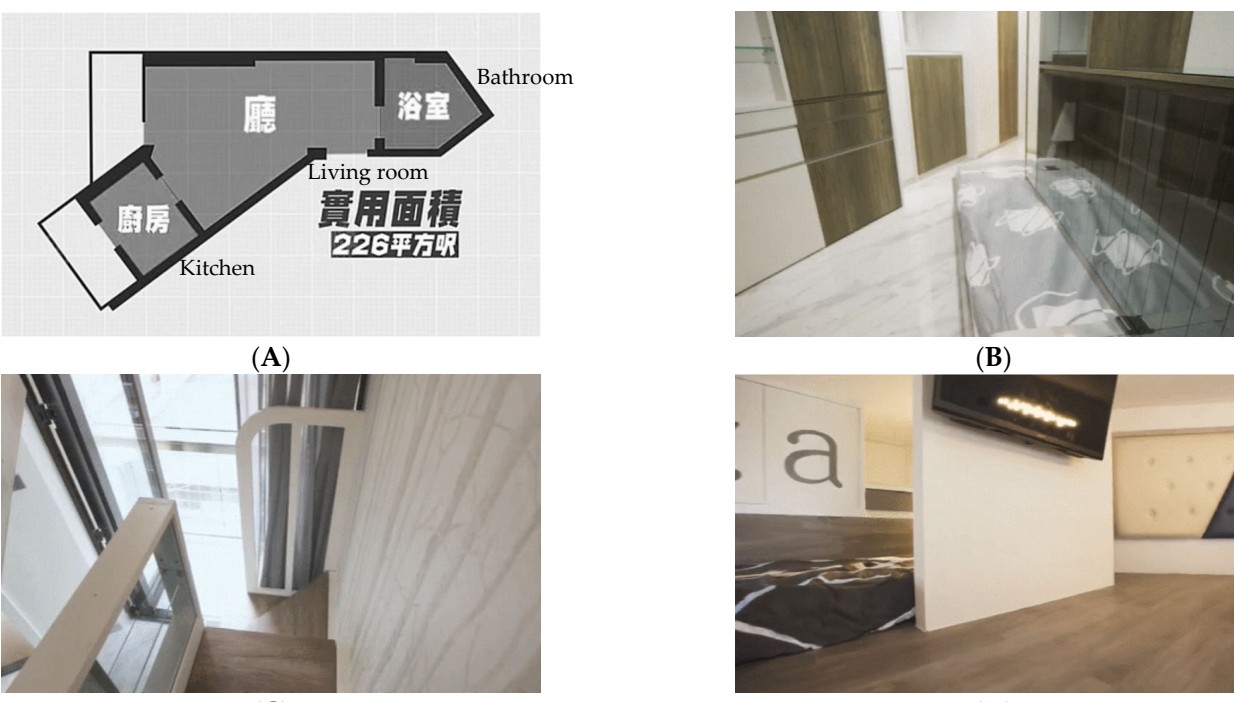

(**A**)  (**B**)

(**C**)  (**D**)

**Figure 13.** A nano flat in Hong Kong. (**A**) Layout of the 15 m² flat. (**B**) Fold-out bed. (**C**) Stairs with cupboard. (**D**) Upper-level bedroom spaces for parents and a child. Source: A family of four people in a 200-foot nano building. Available online: https://www.youtube.com/watch?v=hnF7yGBB1WE (accessed on 7 December 2023) [62].

## 4. Conclusions

While it is common for volume house builders to use a range of standard dwelling types, especially when there are consistent space standards and market expectations [63], the public housing standardisation in Hong Kong is exceptionally far-reaching. Standard block designs were effective in controlling housing design outcomes until this inflexible design approach was challenged by site constraints. The shift to site-specific block and modular dwelling layouts created the greater flexibility needed at the building and site scales and has the potential ability to respond to different housing requirements through MFD. The approach to housing design regulation found in Hong Kong is only possible as the government is the main public housing provider, but an increasing involvement of the private sector means a need for more formal housing design requirements, as evident from the case of MFD.

As the study shows, many dwellings in Hong Kong are too small to fully meet the everyday needs of their users, requiring great levels of innovation and compromise from the occupants. Although there has been a steady increase in the average living space per person in public housing from 2.23 m² in the 1950s to around 13 m² in the late 2010s [23,52], with the actual average living space per person consistently surpassing these standards, this is not solely attributable to effective living space standards. In fact, dwelling sizes have largely remained the same while the average number of occupants decreased from 4.4 persons in 1961 to 2.7 by the end of 2020, leading to an increase in the available space per person [56,64]. This highlights how regulatory approaches and housing standards are

highly contextual. Further studies are needed to understand how suitable dwelling size standards can be better determined and implemented in Hong Kong.

The changing layout of public housing units over time reflects the increase in dwelling size in response to economic and social changes but also shows an increasing articulation of the façade to optimise natural light and ventilation (Figure 14). While issues of housing quality and quality of life directly relate to dwelling size and environmental comfort, the paper has discussed some of the coping strategies implemented by users. However, there remains a significant gap in qualitative research around questions of housing quality, use of the home, lived experience, and social and cultural norms that shape housing design and expectations, as well as perceptions of well-being. While occupants have employed various inventive ways to make small spaces work for them, additional research is needed to understand cultural expectations regarding the home's use and how actual usage patterns in Hong Kong differ from other places.

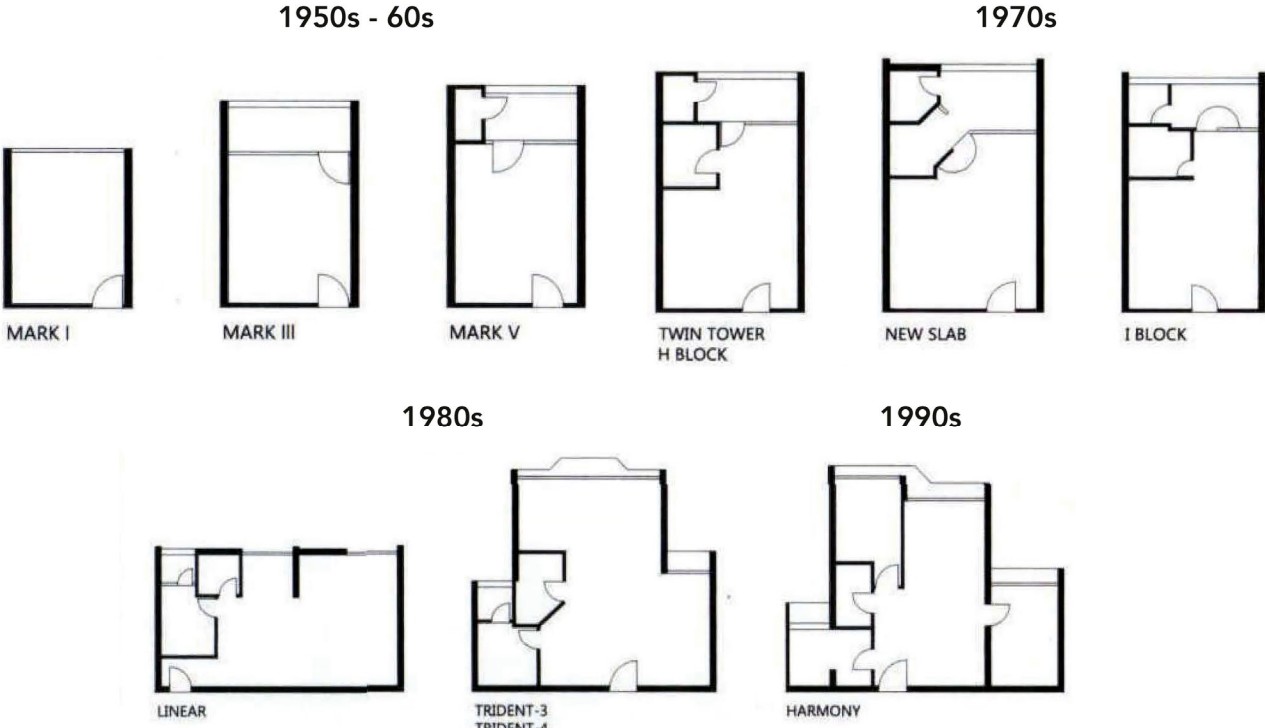

**Figure 14.** Public housing units in typical standard block designs over time.

**Author Contributions:** Conceptualisation, J.C.C., L.W. and S.J.; methodology, J.C.C., L.W. and S.J.; investigation, J.C.C., W.M., L.W. and S.J.; writing—original draft preparation, J.C.C., W.M. and L.W.; writing—review and editing, L.W. and S.J.; visualisation, J.C.C., W.M. and L.W.; funding acquisition, S.J. and L.W. All authors have read and agreed to the published version of the manuscript.

**Funding:** This work was supported by the Prosit Philosophiae Foundation and the China Scholarship Council, grant number 202206155005.

**Institutional Review Board Statement:** Not applicable.

**Informed Consent Statement:** Not applicable.

**Data Availability Statement:** No new data were created or analysed in this study. Data sharing is not applicable to this article.

**Conflicts of Interest:** The authors declare no conflicts of interest.

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
