# Peer review of "Standard Block and Modular Dwelling Designs in Hong Kong’s Public Housing"

_2673-8945, doi:10.3390/architecture4010007_

Round 1
Reviewer 1 Report
Comments and Suggestions for Authors
The article provides a very well developed historical overview over the problems related to minimum housing standards in Hong Kong. It is a solid study with a strong focus on the variation of square meters per person, some information about other aspects of minimum housing standards and policies, and very little about the social, political and cultural context of the city-state that were determining those variations.
But its strenght - as a coherent typological and historical study - is also its weakness, it renders the article a bit too localistic, like a history that is very much self-explanatory and this means it does not yet achieve its potential to be relevant for other cultural spaces and historical times. I think the article can benefit even from small inserts in which the authors may draw even vague comparisons with other situations, either regional or even global.
The abundance of plans is a proof of an industrious effort to document the subject-matter, but there is more need to understand the local cultural flavour, the everyday life and practices, and the urban setting, rather than the evolution of units. However, I appreciated the use of photography and the illustrations of the general plans (the ”parti”), that are bringing valuable information for the reader. I would say that the article would benefit of any additional information - not sure what is the upper limit allowed by the editors - that can be provided about the housing policy context, for now it is an excellent introduction into a very interesting realm that is still hard to read.
Author Response
Many thanks for taking time to review our paper.
We fully understand the critique and wished that there would more scope to expand the paper and compare this to other regions and periods. However, given the focus of the paper on the typological changes, we believe that these tend to be specific to a particular cultural space and historical time. It would require significant more space and would be a very different study if we tried to make this paper have a more global or ahistorical discussion. However, we have tried to clarify further why this paper might be of interest to the reader in the introduction and have also expanded on the economic aspects of rent and the discussion of affordability, as this seemed to be closest to our existing focus on dwelling size. We hope that this is sufficient a revision to address the concerns of the reviewer.
Reviewer 2 Report
Comments and Suggestions for Authors
This article reveals a rigour research and documentation. It should be published.
Author Response
Many thanks for taking time to review our paper.
Reviewer 3 Report
Comments and Suggestions for Authors
The essay is well constructed and clearly written. A few suggestions would be to 1.) diagram more clearly the information in the tables. They currently are hard to decipher and are relatively small. 2.) include more economic information. How were the flats paid for / what did they cost / how long do they last / how does the cost change over time? and 3.) convince me at the beginning as to why the topic is important.
Comments on the Quality of English LanguageI would only ask to confirm if the essay will be published in British or American english. If the latter, some minor adjustments should be made. Otherwise, the essay is clear.
Author Response
Many thanks for taking time to review our paper.
1) We have laid out the large tables differently to increase the text size and legibility. We trust that when published online, they will be available for view in larger/full size, to make them easily readable.
2) Thank you for your suggestion. We have added economic information that we hope addresses the core questions. We have focused hereby on rents, as their discussion seemed most consistent in economic terms with our focus on dwelling sizes.
3) LL47-55: We have added further information on how the use of standard designs is unique to Hong Kong and China, and therefore an interesting case to study to understand if this regulatory approach is effective to the improvement of housing quality as measured in terms of dwelling size. This is important, as aspects of dwelling size is the most common regulatory control used in most other countries.
4) The paper is British English, we have made the necessary corrections.
Reviewer 4 Report
Comments and Suggestions for Authors
This paper has a well-crafted structure, articulate aims, and meticulous methodology. The clarity in articulating research objectives sets a strong foundation, guiding readers seamlessly through the study's progression. The rigor applied to the methods ensures the reliability and validity of the findings, showcasing a commitment to scientific integrity. The visual apparatus also contributes effectively to support the arguments.
Author Response

(The authors gave the same response as above.)

Reviewer 5 Report
Comments and Suggestions for Authors
I find the article very interesting and engaging. I have no fundamental comments. However, I would like to mention that, probably due to a copy and paste oversight, the following paragraphs are repeated twice in a row:
lines 342-348 in lines 349-356
and
lines 471-476 in lines 477-482.
Author Response
Many thanks for taking time to review our paper. We have deleted the repeated text.